# Tracking tuberculosis control using detailed population health and satellite luminosity data: findings from Kazakhstan

Sultan Muratov[1¤a☯]*, Charles Becker[2¤b☯]

1 Department of Economics, Duke University, Durham, North Carolina, United States of America,
2 Department of Economics, Duke University, Durham, North Carolina, United States of America

☯ These authors contributed equally to this work.
¤a Current Address: Department of Financial Stability and Research, National Bank of Kazakhstan, Astana, Kazakhstan
¤b Current Address: Department of Economics, Duke University, Durham, North Carolina, United States of America
* sultan.muratov@alumni.nu.edu.kz

## Abstract

Following the collapse of the Soviet Union, Kazakhstan, like many other formerly socialist countries, experienced a surge in tuberculosis (TB). Despite the successful efforts of Kazakhstan's Ministry of Health in reducing TB related mortality, analysis of TB risk factors in Kazakhstan remains incomplete. This study takes advantage of detailed district-level population health data, and links TB to the presence of man-made environmental damage from the Semipalatinsk Nuclear Test Site and the desiccation of the Aral Sea. Using both propensity score matching and regression models, along with census and satellite nightlight (SNL) data, we examine TB incidence and prevalence from 2000 to 2018, focusing on regions affected by acute environmental disasters. Areas severely exposed to ionizing radiation have converging TB incidence and prevalence. In contrast, regions most affected by the Aral Sea's desiccation continue to have elevated TB levels. Our results suggest that areas officially recognized as "zones of ecological catastrophe" and "zones of ecological crisis" have about 40% and 30% higher prevalence and incidence of TB, respectively. Further analysis of time trends reveals that the significantly elevated TB incidence in these areas appears to be driven by dynamics during 2000–2012 and is not present in more recent years. TB prevalence in the zones of ecological catastrophe and crisis also demonstrates a converging trend, though relative rates remain much higher.

## Introduction

Tuberculosis is among the world's most virulent communicable diseases, though some regions have a far higher prevalence than others [1]. Tuberculosis has also been a major concern for Kazakhstan where the disease surged after the USSR's

**Data availability statement:** All relevant data are within the manuscript and its Supporting Information files.

**Funding:** SM acknowledges financial support from the Global Challenges Research Fund Tuberculosis Social Research and Policy Network (GCRF TB-SRPN) (https://tb-srpn.org/). The funders had no role in study design, data collection and analysis, decision to publish, or preparation of the manuscript.

**Competing interests:** The authors have declared that no competing interests exist.

dissolution [2]. In 1998, together with the World Health Organization, Kazakhstan's Ministry of Health initiated a comprehensive surveillance program known as Directly Observed Therapy Short Course (DOTS) [3,4]. Mass surveillance commenced in 2000: this year serves as the beginning of our series, and the program adds to our confidence in the quality of diagnoses. It also coincided with a National Tuberculosis Program (NTP) that offered free treatment.

Following the initiation of government programs, Kazakhstan's tuberculosis epidemic has shown marked improvement. One of the earlier studies on this topic was conducted by Favorov et al. [4], who contrast TB mortality in Kazakhstan with that in non-DOTS Uzbekistan and four adjacent Russian provinces (Kurgan, Novosibirsk, Omsk, Orenburg) over the period 1998–2004. They conclude that TB mortality declined by roughly 40 percent in Kazakhstan during this period, while it rose sharply in the Russian regions and remained unchanged in Uzbekistan, albeit from much lower base fatality rates. Unfortunately, data for Kazakhstan are not disaggregated by region.

From 2005 through 2016, average annual declines in TB case notification rates and mortality were a remarkable 8.7% and 17.8%, respectively [5]. Despite this success, TB still poses a significant threat to public health. Kazakhstan has one of the highest TB case notification rates (CNR) in the WHO European region and is among the 30 countries designated as having a high burden of multidrug-resistant tuberculosis (MDR-TB) [5].

Given the epidemic situation of TB in Kazakhstan since its Independence in 1991, it is not surprising that the topic has been extensively addressed. In particular, focus has been on identification of the risk factors associated with TB and MDR-TB (Multidrug-resistant TB). Terlikbayeva et al. [2] examine NTP individual data for 2006–2010, finding that characteristics like recent incarceration history, history of contact with TB patients, and unemployment are significantly correlated with case notification rates (CNR). They also find that around 85% of the patients have an unidentified risk factor at the moment of diagnosis [2].

More recent studies on determinants of TB in Kazakhstan have addressed this issue in greater detail. Hermosilla et al. [6], studying risk factors associated with smear-positive TB patients using NTP data for four provinces during 2012–2014, find that males, those with incarceration history, and alcohol dependence or diabetes mellitus (DM) are at higher risk of smear-positive TB. A different approach is taken by Davis et al. [7], who analyze TB-related risk factors on a matched sample of the general population using the same data from NTP as Hermosilla et al. [6]. They conclude that factors like DM, HIV, tobacco or alcohol use, and incarceration history put individuals at greater risk of developing TB.

These studies notwithstanding, analysis of TB risk factors in Kazakhstan remains incomplete. As Davis et al. [7] indicate, existing studies have focused on people with incarceration history or drug addictions, while other risk factors have received limited attention. In particular, a potential factor is exposure to ionizing radiation. It has been established in the academic literature that high doses of ionizing radiation have immunosuppressive effects [8,9]. In turn, people with compromised immune systems are more likely to contract TB [10].

Sadly, Kazakhstan has an extensive history of exposure to ionizing radiation. Most important is the Semipalatinsk Nuclear Test Site (SNTS), where more than 450 nuclear tests were carried out between 1949 and 1989 [11]. In addition to military testing, various parts of Kazakhstan have experienced 39 "peaceful" nuclear explosions that were mainly conducted for industrial purposes [12]. Unfortunately, there is limited literature concerning prolonged exposure to ionizing radiation from the SNTS on the prevalence or incidence of tuberculosis. An exception is Belozerov et al. [13], who report that the incidence of active forms of tuberculosis depends on the level of radiation exposure. They conclude that *raions* (hereafter *districts*) with radiation exposure over three times natural radiation background have 1.4 times and 2.9 times higher incidence of active forms and extrapulmonary forms of TB, respectively. However, their results appear to be based on a comparison of a single affected district with a distant district whose comparability is not well established. Other literature does not appear to systematically address links between atomic testing and tuberculosis epidemics; we address this gap. Importantly, it is crucial to note that our study of population risk cannot establish a causal relationship between radiation exposure and the TB epidemic. Instead, we seek to explore the TB epidemic in areas with historic radiation exposure, considering the existing evidence.

A second major factor that may influence TB levels and distribution in Kazakhstan is the presence of territories with airborne-related ecological risk. Specifically, southwestern Kazakhstan has been heavily affected by the collapse of the Aral Sea's water level. Excessive use for irrigation purposes of the Amu Darya and Syr Darya rivers that flow into the Aral has resulted in an increase in the salt concentration in water and frequent dust storms from the dried-out sea floor [14]. Desiccation of the Aral is a widely recognized problem, and in the popular press has been linked to a spike of diseases, including tuberculosis. These elevated TB levels in the areas affected by the Aral's desiccation have been well-documented by researchers as well [15,16]. Comprehensive measures taken to address the consequences of the Aral crisis also have somewhat improved the epidemiologic situation. Analysis of the Kazakhstani districts in the vicinity of the Aral Sea during 2009–2011 reveals an overall decline in TB mortality rate [17], while district-level analysis in Uzbekistan's Republic of Karakalpakstan concludes that primary incidence of TB in the region declined by 43.7% over the decade from 2005-2015 [18]. Analysis of the structure of primary disability in the Aral Sea region over 2006–2013 also has identified TB as a main contributing factor among respiratory system diseases [19]. Yet, despite these earlier studies, recent academic literature exploring the tuberculosis epidemic in officially-designated "ecological disaster areas" of Kazakhstan seems to be absent. Although we do not establish a causal relationship between TB prevalence and pulmonary damage caused by living in zones of ecological risk, we offer a more comprehensive and nuanced study of Aral Sea desiccation effects, focusing on those districts of greatest stress.

Given the limitations of prior research, much work remains to fully understand Kazakhstan's TB epidemic and its control. We take advantage of the district-level health data published by the MedInform healthcare organization [20]. This dataset contains population information on crude TB prevalence and incidence during 2000–2018 and to our knowledge has not been used in academic research. Integrating both socio-demographic data from the Bureau of National Statistics of Kazakhstan (QazStat) and satellite nightlight luminosity (SNL) data from DMSP-OLS to capture economic characteristics, we analyze the extent to which exposure to ionizing radiation and extreme airborne pollutants affects the TB epidemic, both in aggregate and at the district level.

Furthermore, our paper introduces a technique that, to our knowledge, has not been widely employed in population health studies. In many countries, particularly those outside the highest income bracket, detailed social and economic time series data at the tertiary level are scarce. Our approach involves constructing district-level satellite night-light data from 1992 onward and integrating decennial census data, allowing us to craft a socio-economic history for regions comprising 25,000–75,000 people. This innovative methodology makes detailed population health assessments more feasible, enabling the identification of specific micro-regions of elevated risk over time by comparing regions with similar luminosity and demographics. By taking these steps, our study contributes to the understanding of Kazakhstan's TB epidemic and establishes a methodological approach that may prove valuable for analyses in a broader global context.

## Data

### MedInform

This project utilizes three datasets. Data on crude prevalence and incidence of tuberculosis, as well as healthcare quality indicators, come from the MedInform dataset. This dataset contains detailed population health and medical data of Kazakhstan at district, *oblast* (hereafter *province*) and national level for 2000–2018. We use district-level data that encompass 160 districts plus 38 distinct cities. The cities included in the sample are all designated as official Kazakhstan localities and are treated as districts. Hereafter, we use "district" and "city" interchangeably. It should also be mentioned that MedInform data contains no individual-level information as all health variables are supplied in aggregated form. Therefore, our study does not require any ethical approvals.

Figs 1,2 below represent maps produced from the MedInform Data Presentation System. Fig 1 displays the crude incidence of all forms of TB in Kazakhstan at the province level in 2000 (14 provinces and 2 major cities), while Fig 2 represents the same variable in the same year for Kyzylorda Province at the district level (location of Kyzylorda Province is shown in Fig 1).

### Bureau of National Statistics of Kazakhstan (QazStat)

Many studies find that factors like economic structure, age, sex, and ethnic composition affect tuberculosis incidence and prevalence [5–7]. For this reason, a successful matching procedure for those districts "treated" with radiation or other environmental pollutants relies heavily on the selection of socio-economic and demographic controls. Data on many of these prospective controls are available from the Bureau of National Statistics of Kazakhstan [21]. Specifically, the demographic statistics page provides district-level age-sex distribution and ethnic composition data [21]. These data are available for 2005–2021; for the purposes of this research, the 2005–2018 subset is used. Data for 2000–2004 are absent at the district level. Consequently, age-sex distributions and ethnic compositions are taken from the 1999 Census of Kazakhstan, and the missing years are linearly interpolated. S1 Appendix provides details of the interpolation procedure.

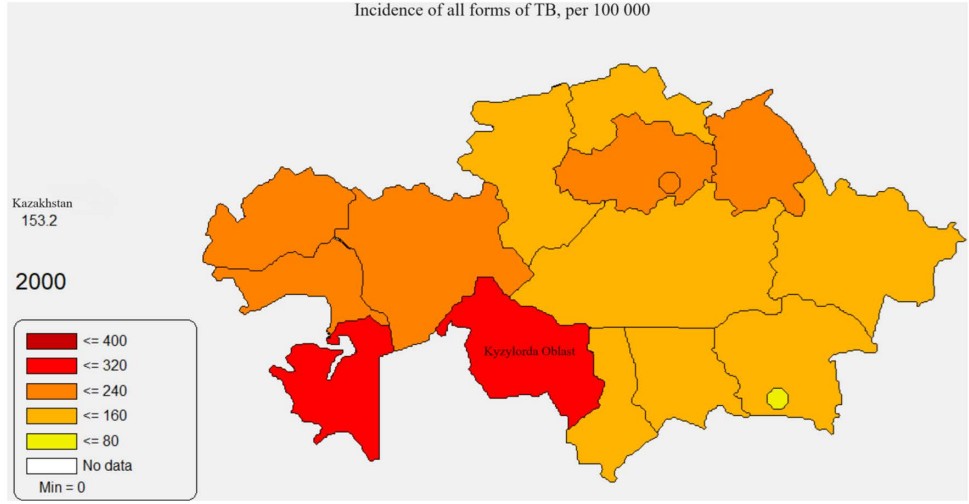

**Fig 1. Incidence of all forms of TB, Kazakhstan, 2000.** Figure 1 was generated on a basis of the MedInform dataset (http://www.medinfo.kz/#/dps-raion), which is a part of a publicly available MedInform DPS program that contains district-level data and allows for a generation of spatial images.

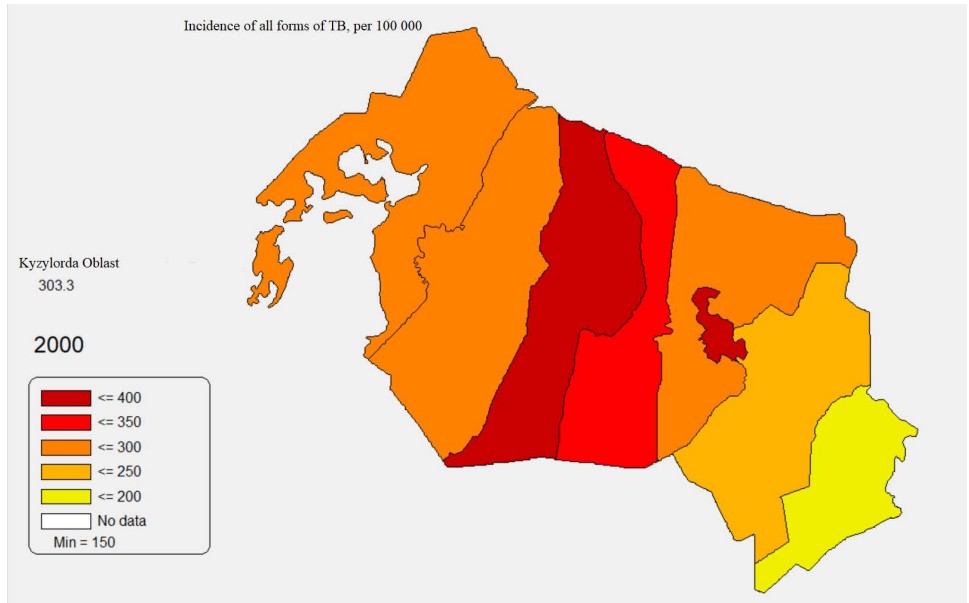

**Fig 2. Incidence of all forms of TB in Kyzylorda Province, 2000.** Figure 2 was generated on a basis of the MedInform dataset (http://www.medinfo.kz/#/dpsraion), which is a part of a publicly available MedInform DPS program that contains district-level data and allows for a generation of spatial images.

### DMSP-OLS

As economic data at the district level are absent for 2000–2018, data from satellite night-light imagery are used as a proxy. Use of satellite nightlight data as a substitute for economic data is not new, and has been extensively applied in existing research [22]. We utilize satellite data from the US Air Force Defense Meteorological Satellite Program (DMSP). This program has focused on collecting visible and near-infrared lights from nighttime human economic activity. The DMSP dataset combines information from six satellites and consists of 34 satellite-years spanning 1992–2013 period [23]. Recently, the dataset was extended with an additional 10 satellite-years to span the 2014–2019 period [24]. From these datasets, we use the "stable lights" products that contain "cleaned up" images of the nighttime world. These images rely on a Digital Number (DN) to represent luminosity on a 0–63 scale, where 63 represents the most luminous parts of the Earth, while 0 represents no lights/background noise. Despite preprocessing, these datasets require the removal of gas flares (which are common in parts of Kazakhstan).

DMSP-OLS satellites lack on-board calibration and the data generate DNs rather than radiance. As a result, comparison of images from different satellites or different years requires an intercalibration process. We adopt the intercalibration method used in Elvidge et al. [25] and described in S2 Appendix to make yearly satellite night lights comparable to one another. Following this procedure, we obtain consistent SNL time-series for 2000–2018 at the district level. The resulting luminosity statistics (i.e., average luminosity, maximum luminosity, etc.) are then utilized as a substitute measure of economic activity of districts.

### "Treated" areas

Defining a treatment group is an integral part of propensity score matching. We determine most of the treatment groups using Kazakhstan government classifications that define supplementary benefit eligibility for those living in/affected by the areas with radiation exposure or environmental pollutants [26]. For the SNTS, Kazakhstan's government offers social

benefits and monetary compensation for people who live in, or who have lived, worked, or completed their military service in territories recognized as "zones of radiation risk". Specifically, these zones are defined as those territories where the dose of the population's exposure exceeded 0.1 rem over the 1949–1990 period. These zones are further divided into four categories depending on the level of radiation exposure: zone of emergency radiation risk (>100 rem), zone of maximal radiation risk (35–100 rem), zone of high radiation risk (7–35 rem), and zone of minimal radiation risk (0.1-7 rem). Because areas that belong to the zone of emergency risk are smaller than districts, and are mostly part of districts in the zone of maximal radiation risk, we merge these two zones together in the analysis that follows. As a result, we utilize three zones of radiation risk: extreme/maximal, high, and minimal.

Apart from zones of radiation risk, Kazakhstan's government has classified other territories as ecological disaster zones. Districts that belong to this group mostly represent the localities affected by the desiccation of the Aral Sea. Depending on the degree of severity of ecological conditions, these territories are split into three categories: zones of ecological catastrophe, ecological crisis zones, and zones of ecological pre-crisis conditions. As these "ecological risk" areas align with district boundaries, we make no changes to the classification of ecological disaster zones.

Unfortunately, mapping the areas of radiation and ecological risk does not complete the determination of treated areas. A complicating factor is that the SNTS was not the only part of Kazakhstan to experience nuclear explosions. As Urgushbaeva et al. [27] detail, underground detonations occurred elsewhere as well: the USSR conducted about 30 "peaceful" nuclear explosions outside of military polygons in Kazakhstan. The main reasons were the opening up of oil or gas fields, capping oil or gas plumes, and accessing other mineral deposits [12]. Since extraction was intended, radiation was especially likely to have escaped. In addition to both peaceful nuclear explosions and nuclear tests for military purposes, Kazakhstan also has many uranium mine tailings, with the greatest concentrations located in Northern Kazakhstan and Almatinskaya provinces. Unfortunately, these other irradiated and polluted areas have not been graded for the level of exposure. However, mapping these areas and excluding them from the matching procedure is crucial. Therefore, we create a single, separate dummy variable *eco_fnd* containing these polluted districts, designating them as "zones of ecological findings." Areas affected by radiation or ecological disasters are presented in Fig 3 below.

## Variables

Our main dependent variables are crude measures of prevalence and incidence of TB aggregated at the district level. The set of controls includes age-sex structure, ethnic composition, and a measure of quality of healthcare services, as well as the level of economic development as represented by satellite night-light data. Our independent variables are characterized by seven dummies for districts that experienced exposure to radiation or environmental pollution. For zones of radiation risk, we create three dummies: ZEMR, ZHR, and ZMR, which represent zones of extreme/maximal, high, and minimal radiation exposure, respectively. 20 districts belong to zones of radiation risk: 3 in ZEMR, 10 in ZHR, and 7 in ZMR. Similarly, we designate four dummies to capture the level of environmental pollution: ECO_CAT, ECO_CRS, ECO_PCR and ECO_FND. These dummies reflect zones of ecological catastrophe, crisis, pre-crisis, and findings, respectively. In total, there are 36 districts in the zones of ecological risk: 3 in ECO_CAT, 6 in ECO_CRS, 11 in ECO_PCR and 16 in ECO_FND. The remaining 142 districts are in the "all-control" group. We provide descriptive statistics of all relevant variables in Table 1 below.

## Methods

### Propensity score matching

Given the lack of a well-defined control group, we adopt a propensity score matching method to define respective controls for zones of radiation and ecological risk. This method is well-suited for observational studies and allows us to match treatment groups to observations as similar as possible based on a list of socio-economic characteristics. Given the

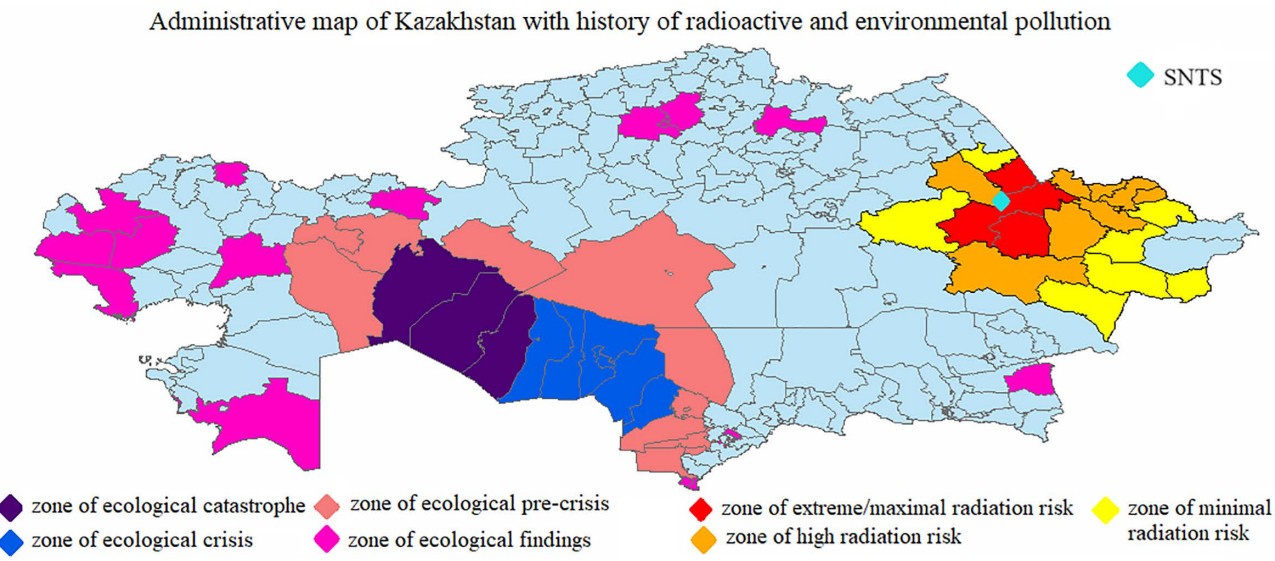

**Fig 3. Administrative map of Kazakhstan with history of radioactive and environmentalpollution.** District-level administrative map of Kazakhstan was obtained from The Humanitarian Data Exchange website (https://data.humdata.org/dataset/kazakhstan-administrative-boundaries-taxonomy).

limited number of districts and the underlying problem that much of Kazakhstan was "treated" in one or more ways, finding comparable, untreated matches is a challenging task. To alleviate some of these issues and improve matching quality, we use the Epanechnikov kernel matching technique with common support. The advantage of this specific matching technique is that more information can be used in the estimation of counterfactual outcomes because kernel matching utilizes weighted averages from all observations in the control group [28]. In addition to selection of the matching technique, PSM also relies on a choice of controls. We balance the districts based on age-sex structure, nationality composition, economic development, and healthcare quality. Unfortunately, due to district heterogeneity in socio-demographic characteristics, it is not possible to use the same set of controls for both radiation and ecological risk zones. However, we carefully interchange the controls to have a representative selection of variables from each of the four categories. Table 2 summarizes the set of PSM controls for radiation and ecological risk zones.

## Regression analysis

To complement PSM and account for time trends, we run a year fixed-effects regression model to better understand the relationship between TB prevalence/incidence and exposure to radiation and environmental catastrophes. We cluster standard errors at the district level to overcome the presence of serial correlation. The detailed regression model is presented below.

$$TB\ Prevalence/Incidence_{it}$$
$$= \alpha + \beta_1 radiation\ exposure\ zone_i + \beta_2 environmental\ disaster\ zone_i$$
$$+ \beta_3 age\ structure_{it} + \beta_4 economic\ development\ level_{it}$$
$$+ \beta_5 nationality\ structure_{it} + \beta_5 year + \varepsilon_{it}$$

where $i$ is district or city, and $t$ is year. The control variables are as follows: *radiation exposure zone* is a vector of binary variables indicating exposure to different levels of radiation ("no exposure" is the omitted term). Correspondingly, *environmental disaster zone* is a vector of binary variables reflecting presence and severity of other environmental disasters

**Table 1. Descriptive statistics of dependent and control variables.**

| | All-control | | ZEMR | | ZHR | | ZMR | | ECO_CAT | | ECO_CRS | | ECO_PCR | | ECO_FND | |
|---|---|---|---|---|---|---|---|---|---|---|---|---|---|---|---|---|
| | N = 2696–2698 | | N = 57 | | N = 190 | | N = 133 | | N = 57 | | N = 114 | | N = 209 | | N = 304 | |
| | Mean | SD | Mean | SD | Mean | SD | Mean | SD | Mean | SD | Mean | SD | Mean | SD | Mean | SD |
| Prevalence of TB | 261.30 | 184.50 | 276.00 | 164.24 | 233.57 | 171.45 | 238.49 | 181.61 | 363.27 | 223.52 | 364.48 | 248.28 | 259.29 | 188.75 | 270.51 | 244.09 |
| Incidence of TB | 111.19 | 61.15 | 124.42 | 64.38 | 107.38 | 50.78 | 102.14 | 54.45 | 141.52 | 90.20 | 144.80 | 90.54 | 107.25 | 62.49 | 103.75 | 57.22 |
| *Economic development: luminosity characteristics* | | | | | | | | | | | | | | | | |
| MEAN | 5.46 | 11.74 | 0.38 | 0.32 | 6.01 | 12.64 | 0.26 | 0.27 | 0.07 | 0.03 | 0.77 | 1.55 | 1.61 | 4.55 | 3.57 | 7.38 |
| STD | 4.62 | 5.00 | 2.35 | 1.62 | 5.21 | 4.03 | 1.67 | 1.04 | 1.07 | 0.23 | 2.76 | 3.60 | 2.54 | 3.60 | 3.82 | 5.35 |
| MAX | 38.31 | 15.69 | 38.58 | 18.02 | 44.35 | 12.27 | 30.82 | 13.72 | 42.81 | 5.10 | 37.83 | 15.52 | 33.11 | 14.83 | 33.89 | 15.46 |
| SUM | 10934 | 12447 | 17162 | 19701 | 12567 | 7065 | 6678 | 5363 | 5896 | 1907 | 6540 | 5654 | 7621 | 8707 | 7160 | 6215 |
| VARIETY | 31.62 | 14.77 | 33.07 | 18.29 | 37.66 | 11.65 | 25.59 | 13.08 | 35.93 | 5.47 | 32.21 | 15.22 | 27.86 | 14.45 | 26.95 | 15.27 |
| *Healthcare quality* | | | | | | | | | | | | | | | | |
| Physicians per 10,000 | 20.29 | 19.85 | 33.91 | 19.08 | 25.67 | 25.48 | 19.22 | 4.80 | 20.76 | 3.85 | 24.81 | 13.94 | 18.47 | 3.10 | 18.10 | 10.23 |
| Nurses per 10,000 | 65.65 | 24.53 | 68.53 | 24.26 | 71.12 | 22.10 | 67.27 | 22.10 | 81.68 | 21.14 | 99.34 | 27.09 | 65.38 | 12.16 | 67.65 | 20.78 |
| *Nationality structure (%)* | | | | | | | | | | | | | | | | |
| % kazakh | 61 | 25 | 76 | 16 | 48 | 31 | 79 | 26 | 99 | 1 | 97 | 3 | 89 | 14 | 72 | 25 |
| % russian | 24 | 18 | 19 | 13 | 46 | 30 | 18 | 26 | 0.5 | 0.4 | 1 | 2 | 2 | 2 | 14 | 17 |
| % ukranian | 5 | 5 | 0.6 | 0.5 | 1 | 0.6 | 0.5 | 0.6 | 0.02 | 0.02 | 0.05 | 0.07 | 0.3 | 0.6 | 2 | 3 |
| *Age-sex structure (%)* | | | | | | | | | | | | | | | | |
| % 0–15 males | 14 | 3 | 13 | 2 | 12 | 2 | 14 | 2 | 17 | 1 | 17 | 1 | 18 | 2 | 16 | 3 |
| % 0–15 females | 13 | 3 | 13 | 2 | 11 | 2 | 13 | 2 | 16 | 1 | 17 | 1 | 17 | 2 | 15 | 3 |
| % 16–62 males | 32 | 2 | 32 | 0.7 | 33 | 1 | 32 | 1 | 31 | 1 | 31 | 1 | 30 | 2 | 31 | 2 |
| % 63 + males | 3 | 0.9 | 4 | 0.7 | 4 | 1 | 4 | 0.9 | 3 | 0.2 | 2 | 0.3 | 2 | 0.7 | 3 | 1 |
| % 58 + females | 8 | 2 | 9 | 2 | 10 | 3 | 8 | 3 | 5 | 0.6 | 5 | 0.7 | 5 | 1 | 7 | 2 |
| % 0–15 total | 27 | 5 | 26 | 4 | 23 | 5 | 27 | 5 | 34 | 3 | 34 | 3 | 35 | 4 | 30 | 6 |
| % 63/58 + total | 11 | 3 | 13 | 2 | 14 | 4 | 12 | 4 | 8 | 0.7 | 7 | 1 | 7 | 2 | 9 | 4 |
| % male | 49 | 2 | 48 | 2 | 49 | 2 | 50 | 1 | 50 | 0.2 | 50 | 0.9 | 50 | 1 | 50 | 1 |

Note: District-level data spans 2000–2018 period. Data on prevalence/incidence of TB as well as healthcare quality come from MedInform. Data on age-sex composition and nationality structure are taken from QazStat, while economic development proxies are derived from DMSP-OLS. MEAN is the mean of night-light luminosity aggregated at district level. STD (standard deviation), MAX, SUM are interpreted analogously to MEAN. VARIETY reflects the number of unique digital numbers for each district.

("no exposure" is the omitted term). Similar to PSM, we control for population structure and economic development levels. These terms absorb unobserved heterogeneity that could be correlated with variables of interest and outcomes, but there is no causal implication. *Age structure* is a vector of terms reflecting the district's population age composition (shares of

children, working age, and retirees; we also control for sex composition), and *nationality structure* is a vector of terms reflecting the district's population ethnic composition. Finally, since we do not have direct measures of economic structure or per capita incomes, we use luminosity (night light) characteristics as our measures of *economic development level*. We opt for the year fixed-effects model above because year dummies allow us to analyze the TB epidemic over time, while treatment zone dummies capture the difference in TB prevalence/incidence between "treated" and "no exposure" areas.

The list of regression controls differs from PSM in three ways. First, regressions are less sensitive to the number of controls. In PSM, the more variables we control for, the harder it is for the algorithm to find adequate matches. Regressions, on the other hand, impose fewer restrictions in this regard, enabling us to use more controls. Second, we do not use healthcare quality variables in regressions, as districts with better healthcare may have better TB detectability, thereby biasing coefficient estimates. We are less concerned with endogeneity issues in PSM, because we want to match districts based on healthcare quality. Lastly, we drop the "percent Kazakh" variable, which represents the share of the Kazakh population of the district. The reasoning is similar to the omission of the healthcare quality indicators. In PSM, we want to ensure matching based on nationality structure, while in regression analysis, we use "percent Kazakh" as a "base" category. The full set of regression controls also appears in Table 2.

## Limitations

As discussed, we control for a wide variety of population characteristics in both the matching and regression models. However, many of these characteristics are based on census data, which means that we cannot fully control for changes in population structure over time. This is especially important since we have only grouped regional prevalence and incidence estimates of TB, and do not have disaggregated age/sex/nationality/urban-rural estimates. Since we do not know annual compositional shifts, it is possible (though not obvious) that some of these terms are correlated with variables we do include, thereby biasing coefficient estimates. Also, it should be mentioned that satellite night-light data, even after the pre-processing and intercalibration procedures, only serve as a proxy for economic development, and do not perfectly correlate with it.

Other limitations include a lack of detailed information on healthcare quality, and an inability to control for the myriad of national and region-specific healthcare policy measures. These effects will be correlated with year and region terms. Consequently, conclusions related to time trends should be understood to also include the effects of public health efforts.

Finally, our analysis does not aim to establish any causal relationship between exposure to environmental hazards from the SNTS and the Aral Sea and TB prevalence/incidence. Identifying causal link would require more detailed data as well as more nuanced econometric techniques that establish an absence of potentially confounding factors. For example, it is conceivable – though unlikely – that public policy sought to remove people from environmentally-damaged areas, and in consequence, there was intentional underinvestment in healthcare.

## Results

### Zones of radiation risk – PSM

This section contains propensity score matching results for prevalence and incidence of TB in zones of radiation risk. The comparison is made across the mean values of treatment, "all-control", and "matched control" groups. The "all-control" group contains all district-level observations excluding areas that were exposed to radiation or ecological disasters. "Matched controls" are calculated in the process of propensity score matching and represent the values of treated observations as if they were untreated. The means of "matched controls" are then compared with the means of treatment groups and used in calculation of average treatment effect on the treated (ATT). ATTs and their p-values after matching are presented in Tables 3,4. Unless noted, all the results are significant at 10% significance level or less. We emphasize that the control group is the same for all treatment groups, as the kernel matching technique relies on weighing all observations in the control group. Hence, means of the control groups before matching coincide.

**Table 2. Set of controls for PSM and Regressions.**

| Variable | PSM SNTS | PSM ECO | Regressions | Source |
|---|---|---|---|---|
| *Economic development: luminosity characteristics* [a] | | | | |
| Mean luminosity (MEAN) | + | + | + | DMSP-OLS |
| Standard deviation of luminosity (STD) | + | – | + | |
| Maximum luminosity (MAX) | – | – | + | |
| Sum of luminosity (SUM) | – | – | + | |
| Variety of luminosity (VARIETY) | + | – | – | |
| *Healthcare quality* | | | | |
| Number of nurses per 10,000 | + | = | = | MedInform |
| Number of physicians per 10,000 | = | + | = | |
| *Nationality Structure* | | | | |
| % kazakh | + | + | – | QazStat |
| % russian | + | + | + | |
| % ukrainian | – | – | + | |
| *Age-sex structure* | | | | |
| % 0–15 males | + | – | + | QazStat |
| % 0–15 females | + | – | + | |
| % 16–62 males | + | – | + | |
| % 63 + males | + | – | + | |
| % 58 + females | + | – | + | |
| % 0–15 total | – | + | – | |
| % 63/58 + total | – | + | – | |
| % males | – | + | – | |

[a] In order to account for heterogenous characteristics of the districts, we utilize several luminosity variables where possible. In principle, use of single luminosity metric, such as SUM, can be enough as such specifications of PSM and regressions produce similar results

Table 3 displays the matching outcomes for the zones of radiation risk. Prior to propensity score matching, the mean prevalence and incidence of tuberculosis in the "treated" districts do not surpass those in the "all-control" group. Additionally, prevalence in high radiation risk zone and incidence in minimal radiation risk zone are statistically lower than the corresponding levels in the "all-control" group. However, after matching districts on their characteristics using kernel PSM, all treatment groups exhibit higher TB prevalence compared to their matched controls. Specifically, we observe a prevalence of 0.28% (treatment) versus 0.18% (matched control) in extreme/maximal, 0.24% versus 0.21% in high, and 0.24% versus 0.20% in minimal radiation risk zones. In other words, prevalence of TB is 1.17-1.60 times higher in treated areas than in their respective controls, though results for the high-risk zone are slightly outside the 10% significance level. Similarly, significant positive mean differences in TB incidence are only observed after matching. Incidence of TB is nearly 0.13% in extreme/maximal, 0.11% in high, and 0.10% in minimal radiation risk zones. These rates stand in contrast to a 0.09% incidence in the matched control group, implying that incidence of TB is 1.13-1.38 times higher in treated areas than in their respective controls. Additional details on matching quality are available in S3 Appendix.

## Zones of ecological risk – PSM

PSM results for the zones of environmental risk appear in Table 4. PSM generates positive differences for two ecological zones with most pronounced effects: catastrophe and crisis. Unlike zones of radiation risk, statistically significant differences are observed both before and after the matching procedure. PSM suggests that prevalence of TB in matched control groups of ecological catastrophe and crisis zones is 0.25% and 0.29%, respectively. This, compared to the 0.36%

**Table 3. PSM results for Zones of radiation risk.**

| Zone of extreme/maximal radiation risk (ZEMR) | Before Propensity Score Matching | | | | After Propensity Score Matching | | | |
|---|---|---|---|---|---|---|---|---|
| | Treatment group (n=57) | All-Control group (n=2696) | Differ-ence | P-value | Treatment group (n=53) | Matched Control group (n=2696) | Differ-ence | P-value |
| Prevalence of all forms of TB | 276.00 | 261.41 | 14.59 | 0.55 | 281.97 | 175.71 | 106.26 | 0.037 |
| Incidence of all forms of TB | 124.42 | 111.22 | 13.20 | 0.11 | 126.31 | 91.58 | 34.73 | 0.047 |
| Zone of high radiation risk (ZHR) | Treatment group (n=190) | All-Control group (n=2696) | Differ-ence | P-value | Treatment group (n=118) | Matched Control group (n=2696) | Differ-ence | P-value |
| Prevalence of all forms of TB | 233.58 | 261.41 | −27.83 | 0.04 | 241.15 | 206.40 | 34.74 | 0.11 |
| Incidence of all forms of TB | 107.38 | 111.22 | −3.84 | 0.40 | 108.03 | 95.31 | 12.72 | 0.05 |
| Zone of minimal radiation risk (ZMR) | Treatment group (n=133) | All-Control group (n=2696) | Differ-ence | P-value | Treatment group (n=131) | Matched Control group (n=2696) | Differ-ence | P-value |
| Prevalence of all forms of TB | 238.49 | 261.41 | −22.91 | 0.16 | 239.75 | 199.79 | 39.96 | 0.04 |
| Incidence of all forms of TB | 102.14 | 111.22 | −9.07 | 0.09 | 102.51 | 89.12 | 13.39 | 0.03 |

**Table 4. PSM results for Zones of ecological risk.**

| Zone of ecological catastrophe (eco_cat) | Before Propensity Score Matching | | | | After Propensity Score Matching | | | |
|---|---|---|---|---|---|---|---|---|
| | Treatment group (n=57) | All-Control group (n=2698) | Differ-ence | P-value | Treatment group (n=57) | Matched Control group (n=2698) | Differ-ence | P-value |
| Prevalence of all forms of TB | 363.27 | 261.30 | 101.97 | <0.001 | 363.27 | 247.25 | 116.02 | 0.004 |
| Incidence of all forms of TB | 141.52 | 111.19 | 30.33 | <0.001 | 141.52 | 112.55 | 28.97 | 0.05 |
| Zone of ecological crisis (eco_crs) | Treatment group (n=114) | All-Control group (n=2698) | Differ-ence | P-value | Treatment group (n=114) | Matched Control group (n=2698) | Differ-ence | P-value |
| Prevalence of all forms of TB | 364.48 | 261.30 | 103.18 | <0.001 | 364.48 | 281.66 | 82.81 | 0.001 |
| Incidence of all forms of TB | 144.80 | 111.19 | 33.61 | <0.001 | 144.80 | 119.53 | 25.27 | 0.005 |
| Zone of ecological pre-crisis (eco_pcr) | Treatment group (n=209) | All-Control group (n=2698) | Differ-ence | P-value | Treatment group (n=192) | Matched Control group (n=2698) | Differ-ence | P-value |
| Prevalence of all forms of TB | 259.29 | 261.30 | −2.01 | 0.88 | 267.31 | 267.06 | 0.25 | 0.99 |
| Incidence of all forms of TB | 107.25 | 111.19 | −3.94 | 0.37 | 109.30 | 116.49 | −7.19 | 0.17 |
| Zone of ecological findings (eco_fnd) | Treatment group (n=304) | All-Control group (n=2698) | Differ-ence | P-value | Treatment group (n=294) | Matched Control group (n=2698) | Differ-ence | P-value |
| Prevalence of all forms of TB | 270.51 | 261.30 | 9.21 | 0.43 | 276.54 | 259.69 | 16.84 | 0.26 |
| Incidence of all forms of TB | 103.75 | 111.19 | −7.44 | 0.04 | 105.54 | 111.68 | −6.14 | 0.09 |

prevalence in both treatment groups, implies that prevalence of TB is 1.3-1.5 times higher in these two zones of ecological risk. As for TB incidence, mean values of two treatment groups are around 0.14% while mean values of the matched controls range from 0.11% (ecological catastrophe) to 0.12% (ecological crisis). In other words, PSM predicts that incidence of TB is about 21–26% higher in zones of ecological catastrophe and crisis than in their matched controls. The results for the zone of ecological pre-crisis are mostly statistically insignificant, while the zone of ecological findings is predicted to have 5% lower incidence of TB than its matched control (details on matching quality appear in S3 Appendix).

### Zones of radiation risk – regression analysis

Summary regression results for prevalence and incidence of TB appear in Table 5 (S4 Appendix provides full regression tables). They reveal contradictory estimates on the relationship between radiation exposure and TB. Specifically, all radiation exposure dummies are statistically insignificant for both prevalence and incidence. In other words, regression results suggest that exposed areas are not very different from the districts in the all-control group. In contrast, PSM results yield positive, significant differences for all treated groups. Differences in the results arise from the weighting procedure in the kernel PSM. For radiation zones, we note that observations in the control group with higher prevalence/incidence values are not necessarily assigned higher weights, which in turn affects the means of "matched controls" and group differences. This is a somewhat puzzling finding, and we find it difficult to clearly conclude with a preference for one method over the other. We believe that this discrepancy can be further addressed as more data become available.

While PSM and regression results are inconsistent with one another, we are somewhat inclined to favor the PSM findings. Our preference reflects the inconsistency in regression results along with the possibility that the underlying regression coefficients are not stable across the full range of observations, while the small number of districts limits our ability to estimate more sophisticated, flexible equations.

### Zones of ecological risk – regression analysis

Table 5 also summarizes regression results for prevalence and incidence of TB in zones of ecological risk. Unlike the zones of radiation risk, our regression specification predicts positive and significant differences for the zones of ecological catastrophe and crisis. To illustrate, the year-FE model with clustered standard errors predicts 102 and 128 additional prevalence cases of TB in zones of ecological catastrophe and crisis, respectively. Compared to the mean for the all-control group, which is equal to 261, this model suggests that these two zones of ecological risk have about 1.4-1.5 times higher prevalence of tuberculosis. Similarly, the same year FE model predicts an additional 38 incidence cases of TB on average in the zones of ecological crisis. As the mean incidence of TB in the all-control group is around 111, we conclude that the zone of ecological crisis has a 34% higher incidence. In contrast to PSM, regression results for incidence of TB in the zone of ecological catastrophe predict a positive yet statistically insignificant difference. Coefficients for the zones of ecological pre-crisis and findings suggest that mean incidence of TB in these regions is comparable to national levels. In general, regression findings for the zones of ecological risk tend to align well with the results from propensity score matching.

### Sensitivity analysis

Additionally, we run a sensitivity check for PSM and regression results outlined above. Given our earlier concerns with endogeneity, we decided to omit the healthcare quality variables (number of nurses per 10,000 and number of physicians per 10,000) from kernel matching and, instead, added them into the regression estimations as controls. The results of this procedure are provided in detail in S5 Appendix. Broadly speaking, sensitivity analysis yields similar results for both PSM and regressions.

**Table 5. The relationship between TB prevalence/incidence and zones of radiation/environmental risk.**

| | Model 1 | Model 2 |
|---|---|---|
| | TB prevalence | TB incidence |
| zemr | −16.94 | 2.554 |
| | (20.53) | (10.06) |
| zhr | −20.41 | −3.825 |
| | (24.24) | (7.162) |
| zmr | −25.96 | −8.127 |
| | (30.22) | (8.439) |
| eco_cat | 101.6** | 30.48 |
| | (46.86) | (19.94) |
| eco_crs | 128.2*** | 37.78*** |
| | (28.14) | (10.46) |
| eco_pcr | 26.89 | 1.791 |
| | (21.21) | (7.518) |
| eco_fnd | 9.327 | −8.672 |
| | (20.47) | (6.363) |
| Constant | 504.2*** | 175.8*** |
| | (99.23) | (39.45) |
| Observations | 3,762 | 3,762 |
| R-squared | 0.630 | 0.542 |
| Year fixed effects | Yes | Yes |
| Cluster SE | Yes | Yes |

Note: zemr, zhr and zmr, represent zones of extreme/maximal, high and minimal radiation exposure, respectively. eco_cat, eco_crs, eco_pcr and eco_fnd reflect zones of ecological catastrophe, crisis, pre-crisis, and findings, respectively.

***, **, and * denote significance at the 1%, 5%, and 10% levels, respectively. Robust standard errors in parentheses.

Most of the changes in the PSM results are observed in the zone of extreme/maximal radiation risk. Kernel matching predicts 143.4 and 75.7 cases of prevalence and incidence of TB in those areas most affected by radiation exposure from SNTS, respectively (versus 175.7 and 91.6 cases of prevalence and incidence of TB in the general regression analysis). Such differences likely arise due to the small sample size of the treatment group and changes in the composition of the matched control group. For regressions, we observe that the addition of healthcare quality controls decreases the magnitude of the coefficients in zones of ecological catastrophe and crisis. In the zone of ecological catastrophe, the updated regression specification predicts only 80.9 additional cases of prevalent TB (versus 101.7 in the general results). As for the zone of ecological crisis, sensitivity analysis predicts only 91.5 and 18.9 additional prevalence and incidence cases of TB, respectively (versus 128.2 and 37.8 in the general results). Although coefficients are smaller in magnitude, they retain their direction and statistical significance.

### Time trends

Finally, it is instructive to analyze the variation of TB prevalence and incidence over time to provide a picture of evolution of TB in "treated" and "control" areas. Fig 4 below summarizes the time trends of these variables. In general, both prevalence and incidence vary with the degree of exposure. Most visible differences come from the zones of extreme/maximal

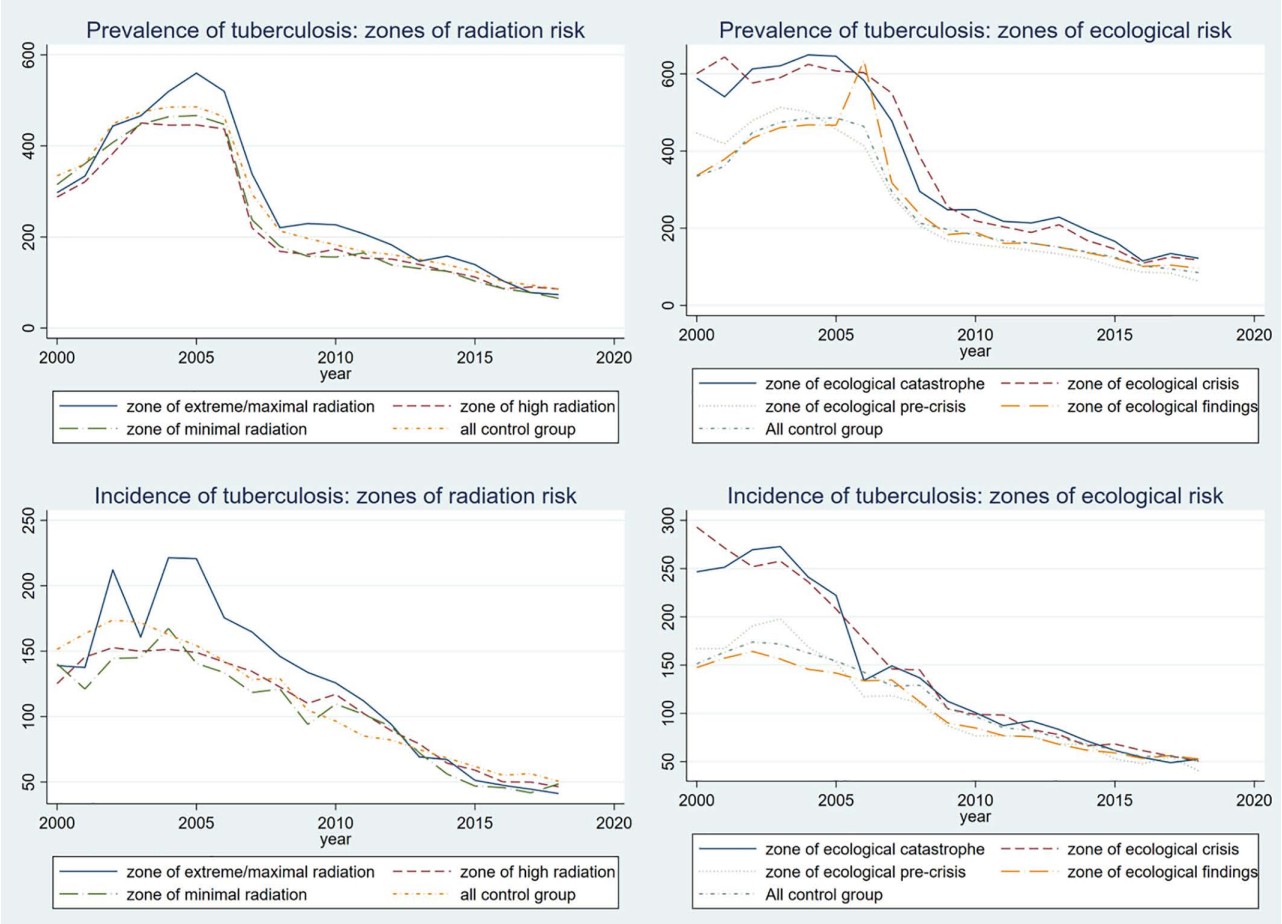

**Fig 4. Time trends of TB prevalence and incidence over 2000-2018 period.**

radiation, and the zones of ecological catastrophe and crisis. Such differences in data persist until 2006–07, after which all of the "treatment" groups converge to the time trend of the all-control group. The trend of the all-control group itself indicates that Kazakhstan's TB epidemic in the country has been in sharp decline throughout 2000–2018.

Motivated by the graphical analysis, we split the time domain into three intervals (2000−05, 2006−12, 2013−18) and run year FE model with clustered standard errors on each of them separately. This procedure allows us to check whether general regressions and PSM patterns observed in the earlier analysis prevail over shorter periods of time. The results of the regression analysis are presented in Table 6.

Table 6 reveals that the results of time interval analysis for TB prevalence have similar significance patterns with those from the complete 2000−18 period. Most of the statistically significant differences are predicted for zones of ecological catastrophe and crisis, and zones of high radiation risk. For zones of ecological risk, all regression models predict positive significant differences for zones of ecological catastrophe and crisis, independent of time interval. Precisely, the zones of ecological catastrophe and crises are expected to have 173 and 186, 99 and 148, and 41 and 50 more prevalence cases over 2000−05, 2006−12, and 2013−18 periods, respectively. The mean in all-control group over the same time intervals is equal to 431, 240, and 116, which indicates that zones of ecological catastrophe/crisis have 1.4, 1.4-1.6, and 1.3-1.4 times higher prevalence over the 2000−05, 2006−12, and 2013−18 periods, respectively. This pattern is one of the key findings:

**Table 6. The relationship between TB prevalence/incidence and zones of radiation/environmental risk by time period.**

| | TB prevalence | | | TB incidence | | |
|---|---|---|---|---|---|---|
| | Model 1 Years 2000–2005 | Model 2 Years 2006-2012 | Model 3 Years 2013-2018 | Model 4 Years 2000-2005 | Model 5 Years 2006-2012 | Model 6 Years 2013-2018 |
| zemr | −30.44 | 13.23 | −10.97 | 10.77 | 17.58 | −10.68 |
| | (48.13) | (21.25) | (10.31) | (17.37) | (11.72) | (6.783) |
| zhr | −11.86 | −35.62* | −15.11 | −3.191 | 0.788 | −8.790* |
| | (58.00) | (19.84) | (9.305) | (16.35) | (6.934) | (4.823) |
| zmr | −29.00 | −30.29 | −16.59 | −16.22 | 1.844 | −10.12** |
| | (55.49) | (28.07) | (14.54) | (17.48) | (9.071) | (4.408) |
| eco_cat | 173.2* | 99.32*** | 40.71* | 84.95** | 11.14 | 2.275 |
| | (91.87) | (35.88) | (24.09) | (33.16) | (19.23) | (9.199) |
| eco_crs | 186.0*** | 148.2*** | 50.16*** | 85.34*** | 20.12* | 8.235 |
| | (40.00) | (38.01) | (13.64) | (19.01) | (11.33) | (5.845) |
| eco_pcr | 67.75 | 21.38 | 2.580 | 14.82 | −6.962 | −0.614 |
| | (44.63) | (19.90) | (9.262) | (17.30) | (7.597) | (3.840) |
| eco_fnd | −13.97 | 36.00 | 3.635 | −15.30 | −7.926 | −1.391 |
| | (34.86) | (28.33) | (8.819) | (11.81) | (6.601) | (4.153) |
| Constant | 660.1*** | 288.2 | −89.46 | 232.5*** | 139.7 | −57.89 |
| | (104.0) | (392.0) | (155.0) | (37.06) | (224.4) | (71.82) |
| Observations | 1,188 | 1,386 | 1,188 | 1,188 | 1,386 | 1,188 |
| R-squared | 0.241 | 0.445 | 0.426 | 0.192 | 0.279 | 0.214 |
| Year FE | Yes | Yes | Yes | Yes | Yes | Yes |
| Cluster SE | Yes | Yes | Yes | Yes | Yes | Yes |

Note: zemr, zhr and zmr, represent zones of extreme/maximal, high and minimal radiation exposure, respectively. eco_cat, eco_crs, eco_pcr and eco_fnd reflect zones of ecological catastrophe, crisis, pre-crisis, and findings, respectively. ***, **, and * denote significance at the 1%, 5%, and 10% levels, respectively. Robust standard errors in parentheses.

despite a decreasing trend in TB prevalence, proportionate differences between the all-control group and zones of ecological catastrophe/crisis are maintained over time. As for zones of radiation risk, the year FE model predicts 36 fewer cases for the zone of high radiation risk in the 2006−12 period. Overall, though, our regression model does not generate strong patterns or consistent values for the radiation risk zones.

As for TB incidence, most zone coefficients are statistically insignificant. Still, some patterns can be observed. From Table 6, our regression model predicts 9−10 fewer incidence cases in zones of high and minimal radiation risk in 2013−18

period. As for the zones of ecological damage, most of the statistical differences are noticed for the zone of ecological crisis. Year FE model predicts around 85 and 20 more incidence cases over the 2000−05 and 2006−12 periods, respectively. Compared to the means for the all-control group, which is equal to 163 and 110 in the respective time intervals, these results imply that TB incidence in ecological crisis zones is 52% higher in 2000−05, and 18% higher in 2006−12. Additionally, the year FE model predicts 85 more incidence cases in the 2000−05 period for the zone of ecological catastrophe. Such differences vanish for later periods.

This time interval analysis provides two potential takeaways. First, for both prevalence and incidence of TB, the regressions predict a clear convergence pattern for most of the treated areas. However, given the absence of age/disease-specific measurements of mortality in our data, it is not possible to determine whether this convergence comes from the improvements in the health care system or deaths of the individuals with TB in "treated" areas – we suspect the former effect is dominant but cannot prove it. Second, in line with our previous findings, the prevalence of TB in zones of ecological catastrophe and crisis is about 1.4 times higher than in the all-control group. However, for TB incidence, only the zone of ecological crisis produces positive significant differences over 2000−05 and 2006−12 periods. These findings indicate that group differences predicted by PSM and general regression analysis for the zone of ecological catastrophe are likely to be caused by variation in the incidence of TB in the initial 2000−05 period. As for the zone of ecological crisis, our findings suggest that the epidemic situation in this area is in decline, but needs to be further monitored. Broadly speaking, incidence converges faster than prevalence, with treatment-resistant strains likely generating this difference.

## Conclusion

Kazakhstan has an extensive history of being exposed to pollution from atomic testing, peaceful nuclear explosions, and environmental disasters. In this study, we investigate the TB epidemic in areas with histories of radiation and environmental pollution. Combining district-level medical information for the 2000−18 period with a set of socio-demographic controls and satellite data, we use both propensity score matching and regression analysis to assess relative risks. Overall, we observe a decline both in prevalence and incidence of TB at the national level. However, we find that areas officially recognized as "zones of ecological catastrophe" and "zones of ecological crisis" have about 40% and 30% higher prevalence and incidence of TB, respectively. We then split the 2000−18 domain into three intervals and repeat the regression analysis to check if our findings prevail over shorter periods. Our detailed analysis confirms that "zones of ecological catastrophe" and "zones of ecological crisis" still have significantly higher TB prevalence.

The results of our study signify the importance of addressing the TB epidemic in areas with environmental and radio-ecological pollution. Despite efforts to include all relevant control variables, our dataset still has areas for improvement. We lack age/disease-specific measurements of mortality, and we proxy economic development indicators with satellite-image data. These gaps can be fulfilled in future studies if more detailed data become available.

We conclude with what is clearly good news: there have been astounding declines in TB incidence and prevalence across Kazakhstan since 2000. Furthermore, there appears to have been convergence across regions even when controlling for different characteristics. Better still, the significantly elevated TB incidence in ecological disaster areas appears to be driven by the early 2000s and is not present in more recent years. There also has been convergence in prevalence of TB in ecological disaster areas, though relative rates remain much higher.

This paper addresses a topic of importance to epidemiologists and public health researchers. Using satellite night light data along with demographic data, we are able to match treated and untreated micro-regions to get a clearer picture of the effects of exposure to the hazards imposed by Aral Sea desiccation and SNTS radiation on TB prevalence and incidence over a long time span as well as shorter periods. These techniques should prove valuable in assessing exposure risk and designing policies in the coming decades, especially in lower- to upper-middle-income countries with good health surveys but erratic or unreliable social and economic data.

## Supporting information

**S1 Appendix. Interpolation of missing years from QazStat.**
(DOCX)

**S2 Appendix. Intercalibration procedure.**
(DOCX)

**S3 Appendix. Details on matching results.**
(DOCX)

**S4 Appendix. Detailed regression tables.**
(DOCX)

**S5 Appendix. Sensitivity analysis results.**
(DOCX)

**S1 File. Supplementary data files.**
(ZIP)

## Acknowledgments

We are grateful to Kateryna Bornukova, Olena Nizalova, Gregory Price, Pavlo Prokopovych, and participants of the GCRF TB Social Research and Policy Network workshops and seminars for valuable comments and assistance.

## Author contributions

**Conceptualization:** Sultan Muratov, Charles Becker.

**Data curation:** Sultan Muratov.

**Formal analysis:** Sultan Muratov, Charles Becker.

**Methodology:** Sultan Muratov, Charles Becker.

**Software:** Charles Becker.

**Supervision:** Charles Becker.

**Validation:** Sultan Muratov.

**Visualization:** Sultan Muratov.

**Writing – original draft:** Sultan Muratov.

**Writing – review & editing:** Charles Becker.

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
