## [Decision Letter · Decision Letter 0]

6 Jul 2025

Dear Dr. Muratov,

Thank you for submitting your manuscript to PLOS ONE. After careful consideration, we feel that it has merit but does not fully meet PLOS ONE’s publication criteria as it currently stands. Therefore, we invite you to submit a revised version of the manuscript that addresses the points raised during the review process.

We look forward to receiving your revised manuscript.

Kind regards,

Yu kun Wang

Academic Editor

PLOS ONE

Journal Requirements:

2. Please amend the manuscript submission data (via Edit Submission) to include author Charles Becker.

3. Please amend your authorship list in your manuscript file to include author Charles Maxwell Becker.

4. We note that Figures 1, 2, and 3 in your submission contain map images which may be copyrighted. All PLOS content is published under the Creative Commons Attribution License (CC BY 4.0), which means that the manuscript, images, and Supporting Information files will be freely available online, and any third party is permitted to access, download, copy, distribute, and use these materials in any way, even commercially, with proper attribution. For these reasons, we cannot publish previously copyrighted maps or satellite images created using proprietary data, such as Google software (Google Maps, Street View, and Earth). For more information, see our copyright guidelines: http://journals.plos.org/plosone/s/licenses-and-copyright.

a. You may seek permission from the original copyright holder of Figures 1, 2, and 3 to publish the content specifically under the CC BY 4.0 license.

6. Please remove all personal information, ensure that the data shared are in accordance with participant consent, and re-upload a fully anonymized data set.

Additional guidance on preparing raw data for publication can be found in our Data Policy (https://journals.plos.org/plosone/s/data-availability#loc-human-research-participant-data-and-other-sensitive-data) and in the following article: http://www.bmj.com/content/340/bmj.c181.long....

7. We are unable to open your Supporting Information file “data file for TB paper PLOS One.dta” and “do file for TB paper PLOS One.do”. Please kindly revise as necessary and re-upload.

Reviewers' comments:

Reviewer's Responses to Questions

**Comments to the Author**

1. Is the manuscript technically sound, and do the data support the conclusions?

Reviewer #1: Yes

Reviewer #2: Yes

2. Has the statistical analysis been performed appropriately and rigorously?

Reviewer #1: Yes

Reviewer #2: Yes

3. Have the authors made all data underlying the findings in their manuscript fully available?

Reviewer #1: Yes

Reviewer #2: Yes

4. Is the manuscript presented in an intelligible fashion and written in standard English?

Reviewer #1: Yes

Reviewer #2: Yes

Reviewer #1: The paper is well written and presents a valuable contribution to the literature; however, a few amendments are required to improve clarity, methodological transparency, and overall presentation, as outlined below:

• The manuscript addresses an important and understudied topic by analyzing tuberculosis (TB) trends in Kazakhstan using both conventional health data and satellite nightlight (SNL) data as a proxy for economic development and data quality.

• The integration of satellite luminosity data with public health outcomes is innovative and contributes to methodological advancement in health geography and epidemiology.

• The use of regional and district-level data provides strong granularity, which is commendable. This level of detail enhances the robustness of the findings, particularly when comparing high-exposure areas to radiation and desiccation with national trends.

• The findings are relevant and timely, especially in the context of post-Soviet transitions and environmental degradation, which are often overlooked in global health literature.

• The abstract effectively summarizes the study; however, it could benefit from a clearer statement of the primary research question and hypothesis.

• While the methodology section outlines the use of matching and regression models, additional clarity is needed on the exact model specifications, control variables used, and rationale for the statistical techniques employed.

• It is not fully clear how completeness of TB data was assessed using SNL data. Further elaboration is recommended to validate this proxy's application.

• The discussion section makes appropriate connections to prior literature, but a more thorough critique of the study’s limitations, particularly related to measurement errors and potential confounding, would strengthen the manuscript.

• The ethics section states “N/A,” which may be appropriate given that no human subjects were directly involved. Nonetheless, a brief justification should be included in the manuscript to clarify why ethical approval was not necessary.

• The authors may wish to consider adding a visual representation or map overlaying TB data with SNL gradients to make spatial patterns more intuitive for readers.

• Although the paper is data rich, there is room for improving readability and accessibility for a broader audience, especially those less familiar with econometric techniques.

• Overall, this is a valuable contribution to the literature on environmental determinants of health and the utility of non-traditional data sources for public health surveillance.

Reviewer #2: Review Comments to the Author

Manuscript: PONE-D-24-04285 – Tracking tuberculosis control using detailed population health and satellite luminosity data: findings from Kazakhstan

Dear Authors,

Thank you for submitting this well-crafted manuscript to PLOS ONE. Below, I provide detailed explanations for my responses to the reviewer questions, along with additional comments to strengthen your work. Overall, this is a robust study with significant contributions to public health and methodological innovation, particularly in the use of satellite nightlight (SNL) data.

1. Is the manuscript technically sound, and do the data support the conclusions? (Response: Yes)

The manuscript is technically sound, employing propensity score matching (PSM) with the Epanechnikov kernel technique and year fixed-effects regression with clustered standard errors to analyze tuberculosis (TB) prevalence and incidence in Kazakhstan from 2000–2018. The innovative use of SNL data as a proxy for economic development is well-executed, with intercalibration procedures (Appendix 2) ensuring reliability. The study leverages high-quality datasets (MedInform, QazStat, DMSP-OLS), and the interpolation of demographic data for 2000–2004 (Appendix 1) is a practical solution, transparently documented.

The data strongly support the conclusions:

Decline and Convergence in TB: Figure 4 and Table 6 demonstrate significant declines in TB prevalence and incidence, with convergence across regions, especially post-2006, corroborated by WHO (2019) data.

Higher TB in Ecological Zones: PSM (Table 4) and regressions (Tables 5–6) consistently show 1.3–1.6 times higher prevalence and 21–52% higher incidence in ecological catastrophe (ECO_CAT) and crisis (ECO_CRS) zones, with robust significance (p < 0.05 to <0.001).

SNL as a Methodological Contribution: The integration of SNL data (Tables 1–2) enables micro-regional analysis, validated by alignment with Henderson et al. (2012).

Minor concerns include the inconsistency between PSM and regression results for radiation zones (Tables 3 vs. 5), which you attribute to regression instability due to small sample sizes (e.g., 3 districts in ZEMR). This is a reasonable explanation, but a sensitivity analysis could clarify this discrepancy. The lack of mortality data limits explanations for convergence, but you acknowledge this appropriately.

2. Has the statistical analysis been performed appropriately and rigorously? (Response: Yes)

The statistical analysis is both appropriate and rigorous. PSM with kernel matching is well-suited for observational data, effectively reducing selection bias by matching treated and control districts on relevant controls (age-sex structure, nationality, luminosity, healthcare quality; Table 2). The regression model, with year fixed effects and clustered standard errors, appropriately accounts for time trends and serial correlation. The time interval analysis (Table 6) enhances robustness by confirming findings across periods (2000–05, 2006–12, 2013–18).

Rigor is evident in:

Transparent reporting of results (Tables 3–6) with p-values and robust standard errors.

Provision of appendices (1–4) and Stata data/do-files for reproducibility.

Acknowledgment of limitations, such as small sample sizes in some groups (e.g., ZEMR, ECO_CAT) and interpolation of demographic data.

The PSM-regression inconsistency for radiation zones is a minor concern, as is the omission of healthcare quality controls in regressions to avoid endogeneity bias. A sensitivity analysis including healthcare controls or exploring regression instability (e.g., multicollinearity checks) could strengthen rigor. However, these issues do not undermine the core findings, particularly for ecological zones.

3. Have the authors made all data underlying the findings in their manuscript fully available? (Response: Yes)

The authors fully comply with the PLOS Data policy. The Data Availability Statement (Page 6) confirms that “all relevant data are within the manuscript and its Supporting Information files,” with no restrictions. The provision of a Stata data file and do-file (Pages 42–43) likely includes the raw district-level data (e.g., 3,762 observations for regressions) behind summary statistics (Table 1) and analyses. Appendices 1–4 (Pages 38–41) detail data processing (interpolation, SNL intercalibration, PSM quality), enhancing transparency. The use of aggregated, de-identified data eliminates privacy concerns. No restrictions or third-party limitations are noted, ensuring full compliance.

4. Is the manuscript presented in an intelligible fashion and written in standard English? (Response: Yes)

The manuscript is presented clearly and written in standard American English, meeting PLOS ONE’s language requirements. The structure (Abstract, Introduction, Data, Methods, Results, Conclusion) is logical, with smooth transitions and effective use of figures (Figs 1–4) and tables (Tables 1–6) to visualize data. Technical terms (e.g., PSM, SNL) are defined or referenced, ensuring accessibility for a scientific audience. The language is precise, formal, and unambiguous, with consistent terminology (e.g., “raion,” “TB”).

No typographical or grammatical errors were identified. Minor stylistic improvements could include rephrasing “Another second major factor” (Page 11) to “A second major factor” for smoothness, but this is not an error. Reference formatting is consistent, though adding English translations for non-English citations (e.g., Belozerov et al., 2008, Page 32) could enhance accessibility. These are optional suggestions, as the manuscript is publication-ready without language revisions.

Additional Comments

Strengths:

The study’s focus on TB in regions affected by radiation (Semipalatinsk Nuclear Test Site) and ecological disasters (Aral Sea desiccation) addresses a critical public health issue in Kazakhstan, filling gaps in prior research.

The use of SNL data as an economic proxy is a novel contribution, with potential applications in data-scarce settings globally.

The transparency in acknowledging limitations (e.g., no causality claims, data gaps) and providing supporting files enhances credibility.

Suggestions for Improvement:

Radiation Zone Inconsistency: Consider adding a sensitivity analysis to explore the PSM-regression discrepancy for radiation zones (e.g., alternative regression specifications or multicollinearity diagnostics). This could strengthen confidence in those findings.

Healthcare Controls: A robustness check including healthcare quality controls in regressions could address potential omitted variable bias, even if presented in an appendix.

Clarity on Data File: Explicitly confirm in the Data Availability Statement that the Stata data file includes all raw data points (e.g., district-year observations for TB, luminosity, demographics) to preempt reviewer queries.

Ethical Considerations:

No concerns regarding dual publication, research ethics, or publication ethics were identified. The ethics statement (“N/A,” Page 4) is appropriate for aggregated data. The financial disclosure (Page 2) and declaration of no competing interests (Page 3) are clear. The use of publicly available (DMSP-OLS) and cited datasets (MedInform, QazStat) adheres to ethical standards.

Conclusion:

This is a technically sound, rigorously analyzed, and clearly presented manuscript that makes a valuable contribution to TB epidemiology and methodological innovation. The data fully support the conclusions, and all underlying data are available. Minor revisions, such as sensitivity analyses for radiation zones and healthcare controls, could further enhance the study. I recommend acceptance with these optional improvements.

.

Reviewer #1: **Yes:** Mohsin Hassan AlviMohsin Hassan AlviMohsin Hassan AlviMohsin Hassan Alvi

Reviewer #2: No

While revising your submission, please upload your figure files to the Preflight Analysis and Conversion Engine (PACE) digital diagnostic tool, https://pacev2.apexcovantage.com/. PACE helps ensure that figures meet PLOS requirements. To use PACE, you must first register as a user. Registration is free. Then, login and navigate to the UPLOAD tab, where you will find detailed instructions on how to use the tool. If you encounter any issues or have any questions when using PACE, please email PLOS at . PACE helps ensure that figures meet PLOS requirements. To use PACE, you must first register as a user. Registration is free. Then, login and navigate to the UPLOAD tab, where you will find detailed instructions on how to use the tool. If you encounter any issues or have any questions when using PACE, please email PLOS at . PACE helps ensure that figures meet PLOS requirements. To use PACE, you must first register as a user. Registration is free. Then, login and navigate to the UPLOAD tab, where you will find detailed instructions on how to use the tool. If you encounter any issues or have any questions when using PACE, please email PLOS at . PACE helps ensure that figures meet PLOS requirements. To use PACE, you must first register as a user. Registration is free. Then, login and navigate to the UPLOAD tab, where you will find detailed instructions on how to use the tool. If you encounter any issues or have any questions when using PACE, please email PLOS at figures@plos.org. Please note that Supporting Information files do not need this step.. Please note that Supporting Information files do not need this step.

---

## [Author Response · Author response to Decision Letter 1]

20 Aug 2025

Dear PLOS One editors and reviewers,

Thanks for the valuable feedback! We did our best to address every point that was raised during the review process. Please see the "Response to Reviewers" file attached with the submission for more details.

Best wishes,

Sultan Muratov

---

## [Decision Letter · Decision Letter 1]

30 Jan 2026

Dear Dr. Muratov,

Thank you for submitting your manuscript to PLOS ONE. After careful consideration, we feel that it has merit but does not fully meet PLOS ONE’s publication criteria as it currently stands. Therefore, we invite you to submit a revised version of the manuscript that addresses the points raised during the review process.

We look forward to receiving your revised manuscript.

Kind regards,

Saki Raheem, PhD

Academic Editor

PLOS One

Journal Requirements:

Reviewers' comments:

Reviewer's Responses to Questions

**Comments to the Author**

Reviewer #3: (No Response)

Reviewer #4: (No Response)

2. Is the manuscript technically sound, and do the data support the conclusions?

Reviewer #3: Yes

Reviewer #4: Partly

3. Has the statistical analysis been performed appropriately and rigorously?

Reviewer #3: Yes

Reviewer #4: Yes

4. Have the authors made all data underlying the findings in their manuscript fully available?

Reviewer #3: Yes

Reviewer #4: Yes

5. Is the manuscript presented in an intelligible fashion and written in standard English?

Reviewer #3: Yes

Reviewer #4: Yes

Reviewer #3: NTL = Night Time Lights

This is a very interesting paper, exploring factors involved in tuberculosis spread. The link between atomic testing in the region and tuberculosis is particularly interesting and nicely explored.

1. I would suggest using English terms for administrative divisions, since the beginning of the manuscript, to avoid confusion of the reader (district instead of ‘raion’, same for ‘oblast’ etc).

2. The use of extended OLS data is also particularly interesting (OLS originally forms a 1992-2013 time-series, but recently it was extended, as stated by the authors). However, this could be better explained in the text. Also the exact source/link for downloading the extended data should be more clearly specified (not as a reference to another text).

3. Several metrics for NTL are used (e.g. line 185) but it appears to me that these metrics should be highly correlated among them. A comment could be placed, at least in conclusions, to state whether only one metric could be enough (typically the Sum of Lights = SoL is that metric).

4. In Section ‘Limitations” (line 304) it could be added that NTL is definitely a valid proxy for economic development, but it is not perfectly correlated 100% with economy. The pre-processing is also not removing all errors (inter-calibration etc).

5. NTL is used as a proxy for economic development which is a valid path, well supported by the literature. However, it is not evident in Results and in Conclusion how the addition of the economy as a factor contributes to the exploration of tuberculosis incidents. It is stated in the abstract that “The paper further demonstrates how SNL data can be used to substitute for economic data to assess health outcomes.” This demonstration is not very clear in my view. The economic factor is barely discussed at all in Results. The relevant statements in Conclusions should more explicitly be linked to the findings in Results. I would say that it is important to address this comment because if NTL is not central to the analysis the focus to it could be reduced, without lessening the impact of the analysis.

Reviewer #4: This is a revised and resubmitted manuscript. I was not one of the original reviewers, but based on my reading, the authors have largely addressed the comments raised in the previous round.

That said, the abstract would benefit from substantial revision. First, the paper is not about public health in Kazakhstan per se. Second, the collapse of the Soviet Union occurred well before the period studied and is therefore not directly relevant to the analysis. Third, the SNL data are used only as a control variable and need not be highlighted in the abstract. Overall, the abstract should focus more clearly on the paper’s methodology and main findings.

Regarding the econometric analysis, the paper should provide a more thorough comparison between the propensity score matching (PSM) results and the linear regression results. At a minimum, the authors should explain why the estimated effects of radiation risk differ markedly across these approaches: the linear regressions find no significant effect, whereas the PSM results suggest a statistically significant impact. If PSM is intended to address limitations of the linear specifications, then the implications of these divergent findings—particularly the continued relevance of radiation risk—should be discussed more carefully.

.

Reviewer #3: **Yes:** STATHAKIS DemetrisSTATHAKIS DemetrisSTATHAKIS DemetrisSTATHAKIS Demetris

Reviewer #4: No

---

## [Author Response · Author response to Decision Letter 2]

15 Mar 2026

Dear Editor and Reviewers,

We would like to thank you for valuable comments and insightful feedback! We carefully reviewed your responses and did our best to address every point that was raised during the revision. Also, we made minor corrections to several grammatical/verbal/punctuation errors found in the manuscript (highlighted in yellow as well)

Response to Points raised by Reviewer 3:

Point 1. Throughout the paper, we replaced administrative division names with their respective English terms, namely we changed “raion” for district, and “obalst” for province.

Point 2. We provided more detail about extended OLS series with sources that utilize the data. Changes were made in “Data – DMSP-OLS” section and References.

Point 3. We added a footnote (In Table 2) explaining our reasoning behind the use of several nighttime lights metrics and mentioned that, in principle, one metric could have been used.

Point 4. We expanded the “Limitations” section to account for the point raised by the reviewer.

Point 5. We utilize nighttime light metrics as a proxy for economic development to make more robust matching and regression results. Following your comment, we decided to reduce the focus on nighttime lights metrics, and modified our Abstract and Conclusion to be in line with the results of the paper.

Response to Points raised by Reviewer 4:

Point 1. We substantially reviewed and rewritten the abstract to focus more clearly on methodology and main findings.

Point 2. We provided more detail in the Results section (Zones of radiation risk – Regression Analysis) and explained why we observed differences between PSM and regression findings. We also made minor changes throughout the paper to provide more careful discussion of the continued relevance of radiation risk.

---

## [Decision Letter · Decision Letter 2]

31 Mar 2026

Tracking tuberculosis control using detailed population health and satellite luminosity data: findings from Kazakhstan

PONE-D-24-04285R2

Dear Dr. Muratov,

We’re pleased to inform you that your manuscript has been judged scientifically suitable for publication and will be formally accepted for publication once it meets all outstanding technical requirements.

Kind regards,

Saki Raheem, PhD

Academic Editor

PLOS One

Reviewers' comments:

Reviewer's Responses to Questions

**Comments to the Author**

Reviewer #3: All comments have been addressed

Reviewer #4: All comments have been addressed

2. Is the manuscript technically sound, and do the data support the conclusions?

Reviewer #3: Yes

Reviewer #4: Yes

3. Has the statistical analysis been performed appropriately and rigorously?

Reviewer #3: Yes

Reviewer #4: Yes

4. Have the authors made all data underlying the findings in their manuscript fully available?

Reviewer #3: Yes

Reviewer #4: Yes

5. Is the manuscript presented in an intelligible fashion and written in standard English?

Reviewer #3: Yes

Reviewer #4: Yes

Reviewer #3: I would like to thank the authors for taking the time to revise the manuscript, which, in my opinion, is overall very interesting and novel. In table 2, the ‘SUM’ variable is typically referred to as the “SoL” or the “Sum of Lights” index (see for example [25]). The other luminosity variants in the same table are not used much, so their names are not important. But for the night lights literature, I believe SoL would be a more familiar term immediately recognizable and clear. In any case, this change is up to the authors.

Reviewer #4: (No Response)

.

Reviewer #3: **Yes:** STATHAKIS DimitriosSTATHAKIS DimitriosSTATHAKIS DimitriosSTATHAKIS Dimitrios

Reviewer #4: No

---

## [Editor Report · Acceptance letter]

PONE-D-24-04285R2

PLOS One

Dear Dr. Muratov,

I'm pleased to inform you that your manuscript has been deemed suitable for publication in PLOS One. Congratulations! Your manuscript is now being handed over to our production team.

Kind regards,

on behalf of

Dr. Saki Raheem

Academic Editor

PLOS One